# Structure and Biosynthetic Gene Cluster of Sulfated Capsular Polysaccharide from the Marine Bacterium *Vibrio* sp. KMM 8419

**DOI:** 10.3390/ijms252312927

**Published:** 2024-12-01

**Authors:** Maxim S. Kokoulin, Yulia V. Savicheva, Nadezhda Y. Otstavnykh, Valeria V. Kurilenko, Dmitry A. Meleshko, Marina P. Isaeva

**Affiliations:** 1G.B. Elyakov Pacific Institute of Bioorganic Chemistry, Far Eastern Branch, Russian Academy of Sciences, 159/2, Prospect 100 let Vladivostoku, Vladivostok 690022, Russia; iu.savicheva0@yandex.ru (Y.V.S.); chernysheva.nadezhda@gmail.com (N.Y.O.); valerie@piboc.dvo.ru (V.V.K.); 2Principal Engineering School, ITMO University, 9, Lomonosova Street, St. Petersburg 191002, Russia; meleshko.dmitrii@gmail.com

**Keywords:** marine bacteria, *Vibrio* sp. KMM 8419, sulfated capsular polysaccharide, NMR, genome, biosynthetic gene cluster

## Abstract

*Vibrio* sp. KMM 8419 (=CB1-14) is a Gram-negative bacterium isolated from a food-net mucus sample of marine polychaete *Chaetopterus cautus* collected in the Sea of Japan. Here, we report the structure and biosynthetic gene cluster of the capsular polysaccharide (CPS) from strain KMM 8419. The CPS was isolated and studied by one- and two-dimensional ^1^H and ^13^C nuclear magnetic resonance (NMR) spectroscopy. The molecular weight of the CPS was about 254 kDa. The CPS consisted of disaccharide repeating units of D-glucose and sulfated and acetylated L-rhamnose established as →2)-α-L-Rha*p*3S4Ac-(1→6)-α-D-Glc*p*-(1→. To identify the genes responsible for CPS biosynthesis, whole-genome sequencing of KMM 8419 was carried out. Based on the genome annotations together with the Interproscan, UniProt and AntiSMASH results, a CPS-related gene cluster of 80 genes was found on chromosome 1. This cluster contained sets of genes encoding for the nucleotide sugar biosynthesis (UDP-Glc and dTDP-Rha), assembly (glycosyltransferases (GT)), transport (ABC transporter) and sulfation (PAPS biosynthesis and sulfotransferases) of the sulfated CPS. A hypothetical model for the assembly and transportation of the sulfated CPS was also proposed. In addition, this locus included genes for O-antigen biosynthesis. Further studies of biological activity, the structure–activity relationship in the new sulfated polysaccharide and its biosynthesis are necessary for the development of potent anticancer agents or drug delivery systems.

## 1. Introduction

Polysaccharides are one of the most diverse biopolymers in nature, explained by the great variety of monosaccharides and the numerous linkage positions between residues [1]. In addition, polysaccharides are often decorated by organic (acetate, lactate, pyruvate, amino acids) and inorganic (phosphate, sulfate) substituents, significantly increasing the number of possible structures. The main interest in the study of polysaccharides is associated with the fact that polysaccharides have numerous interesting properties relevant for potential applications in various industrial sectors, including pharmaceuticals and cosmetics [2,3,4]. Among naturally occurring carbohydrate polymers, exopolysaccharides (EPSs) from bacteria are of increasing scientific interest. Bacterial EPSs can be secreted into the extracellular environment (medium-released polysaccharides, MRPs) or be bound to the cell surface (capsular polysaccharides, CPSs) and represent a great reservoir of new molecules whose structures and functional properties are largely unknown [5]. Indeed, bacterial polysaccharides have a more than ten-fold greater diversity regarding the nature of monomers when compared to glycans from eukaryotic organisms [6]. The structural versatility of bacterial polysaccharides is associated with a broad spectrum of biological functions, such as adhesion; colonization; camouflage; protection against desiccation, toxic compounds and detergents; and others [7,8]. Of particular interest in terms of structure and potential applications are polysaccharides from marine bacteria [9,10,11]. Marine bacteria can live in all marine environments due to a variety of survival strategies, one of which is the ability to produce protective polysaccharides on the surface of their cells. In combination with other biopolymers, they provide a buffered microenvironment in which bacteria can resist stressful conditions and thereby ensure their survival and proliferation [7,8]. In addition, bacteria change phenotype when colonizing new environments. These adaptive modifications are likely accompanied by alterations in the mechanics of the bacterial cell [12]. Investigating the involvement of polysaccharides in these processes can reveal how they help bacteria adapt to their environment and provide insights into the mechanisms that regulate their growth in specific niches. This understanding is meaningful for developing effective antimicrobial chemotherapy.

Despite the numerous polysaccharides identified, the diversity of marine bacterial polysaccharides is mainly underestimated, and it is reasonable to assume that the study of novel natural polysaccharides may provide options for new materials and pharmaceuticals. Among polysaccharide-producing bacteria, the genus *Vibrio* is one of the most common [13]. It is present in marine, estuarine and freshwater systems worldwide and currently comprises 152 validly described species (list of prokaryotic names with standing in nomenclature is available at https://lpsn.dsmz.de/genus/vibrio (accessed on 24 October 2024) [14]). Still, only a handful of EPS structures from *Vibrio* spp. are established to date [15,16,17,18,19,20]. To detect sulfated polysaccharides produced by marine bacteria, a number of *Vibrio* spp. isolated from a food-net mucus sample of marine polychaete *Chaetopterus cautus* (formerly known as *Chaetopterus variopedatus*) collected in the Sea of Japan [21] were analyzed. As a result, only the strain *Vibrio* sp. KMM 8419 demonstrated the ability to produce a sulfated polymer.

The aim of this study was to determine the structure of the sulfated polysaccharide of *Vibrio* sp. KMM 8419 and predict its biosynthesis based on whole-genome sequencing followed by comprehensive bioinformatics analysis.

## 2. Results

### 2.1. CPS Extraction, Purification and General Characterization

The six strains of *Vibrio* spp. (CB 2-11, CB 2-8, CB 2-7, CB 2-1, CB 1-14, CB 2-10) associated with marine polychaete *C. cautus* [21] were analyzed for their ability to produce sulfated polysaccharides. To quickly assess the presence of negatively charged polymers, the crude CPSs were washed from the surface of the bacterial cells and analyzed using toluidine blue-stained electrophoresis in a polyacrylamide gel. Among all the strains examined, only the strain *Vibrio* sp. KMM 8419 exhibited the presence of a sulfated polymer (Appendix A).

The KMM 8419 dried bacterial cells were subjected to extraction with a saline solution followed by enzymatic treatment, and the resulting CPS was purified by hydrophobic interaction chromatography and anion exchange chromatography and desalted by size-exclusion chromatography. The absence of proteins in the pure preparation was concluded by the Bradford assay. The silver-stained SDS-PAGE of the purified CPS revealed no ladder-like bands characteristic of lipopolysaccharide (LPS) contamination, while the toluidine blue-stained SDS-PAGE indicated a broad smear in the upper part of the gel indicative of a negatively charged polysaccharide. The size-exclusion chromatography (SEC) analysis showed that the CPS formed a single symmetrical peak with a molecular weight of about 254 kDa (Figure 1).

Sugar analysis by GC–MS of the acetylated methyl and (S)-2-octyl glycosides revealed the presence of L-rhamnose (L-Rha) and D-glucose (D-Glc). In confirmation of the absence of an LPS impurity, 3-deoxy-D-mannooct-2-ulosonic acid and 3-hydroxy fatty acids were not detected (typical for LPSs). The FT-IR spectrum of the CPS (absorption band at 1230 cm^−1^), in combination with the data from the turbidimetric analysis, indicated the presence of sulfate groups in the CPS.

### 2.2. NMR Analysis of CPS

The ^1^H NMR spectrum of the CPS (Figure 2A) showed three signals in the anomeric region at δ*_H_* 5.04 (H-1 of residue **A**), 4.99 (H-4 of residue **B,** see below) and 4.96 (H-1 of residue **B**) and in the carbinolic region at *δ_H_* 3.59–4.69 (overlapping with HDO from solvent); furthermore, one *O*-acetyl group and one methyl group of 6-deoxysugar at δ*_H_* 2.14 and 1.24 (^3^*J*_H5-H6_ 5.3 Hz, H-6 of residue **B**), respectively, were detected. The ^13^C NMR spectrum of the CPS (Figure 2B) showed, inter alia, two signals in the anomeric region at δ*_C_* 99.6 (C-1 of residue **A**) and 99.1 (C-1 of residue **B**); one substituted hydroxymethyl group at δ*_C_* 67.1 (C-6 of residue **A,** data from DEPT-135 experiment); one methyl group of an *O*-acetyl substituent at *δ*_C_ 21.7; and one methyl group of 6-deoxysugar at *δ*_C_ 17.5 (C-6 of residue **B**).

The absence of signals in the region of δ*_C_* 83–88 demonstrated the pyranose form of all monosaccharide residues. The ^1^*J*_C1-H1_ (≥170 Hz) coupling constant values indicated that both sugar residues were α-linked. Therefore, the CPS consisted of sulfated disaccharide repeating units containing α-L-Rha*p* and α-D-Glc*p* residues.

The repeating unit of the CPS was characterized by the ^1^H,^1^H COSY, ^1^H,^1^H TOCSY and ^1^H,^13^C HSQC experiments (Table 1). In detail, the ^1^H,^1^H COSY and ^1^H,^1^H TOCSY (Figure 3) spectra revealed two isolated proton spin systems (**A** and **B**). Spin system **A,** with a *gluco* configuration, was identified by H-1/H-2 up to H-6 correlations. Spin system **B,** with a *manno* configuration, was revealed by H-1/H-2 and H-2 up to H-6 correlations. The ^1^H,^13^C HSQC experiment (Figure 4) demonstrated two anomeric cross-peaks, numerous sugar ring cross-peaks and methyl group cross-peaks of *O*-acetyl and 6-deoxysugar. The two sugar spin systems were identified as α-D-Glc*p* (**A**) and α-L-Rha*p* (**B**, H-6/C-6 correlation at δ_H_/δ_C_ 1.24/17.5) residues. Thus, the data obtained from the NMR experiments were in accordance with the chemical analysis of the CPS.

The substitution positions and sequences of the sugar residues within the repeating unit of the CPS were established using the ^1^H,^1^H-ROESY and ^1^H,^13^C-HMBC experiments. The following inter-residue cross-peaks between transglycosidic protons were observed in the ^1^H,^1^H-ROESY spectrum (Figure 4A) of the CPS: **A** H-1/**B** H-2 at δ_H_/δ_H_ 5.04/4.30 and **B** H-1/**A** H-6 at δ_H_/δ_H_ 4.96/4.02, 3.86. The ^1^H,^13^C-HMBC experiment (Figure 4B) revealed the respective correlations between the following anomeric protons and the linkage carbons: **A** H-1/**B** C-2 at δ_H_/δ_C_ 5.04/76.2 and **B** H-1/**A** C-6 at δ_H_/δ_C_ 4.96/67.1. In addition, the correlation between H-4 of residue **B** and the *O*-acetyl group at δ_H_/δ_C_ 4.99/174.8 in the ^1^H,^13^C-HMBC experiment indicated the location of the *O*-acetyl substituent at the corresponding position (*O*-4 of α-L-Rha*p*).

A significant downfield displacement of the C-3 signal of residue **B** at δ_C_ 76.2 compared to the chemical shift in the corresponding non-substituted monosaccharide [22] indicated the location of the sulfate group (*O*-3 of α-L-Rha*p*).

Based on the data obtained, the repeating unit of the CPS from *Vibrio* sp. KMM 8419 had the following structure, as shown in Figure 5.

### 2.3. CPS Gene Cluster Prediction

The complete genome of *Vibrio* sp. KMM 8419 was assembled de novo and found to consist of two circular chromosomes (3,497,892 bp and 1,804,652 bp) (Figure 6) and one plasmid (241,015 bp). The genome contains 4778 CDSs, 113 tRNAs and 30 rRNA genes (organized into ten 16S-23S-5S operons). A search for polysaccharide-related genes revealed the genes encoding the biosynthesis of EPS, core oligosaccharide of LPS, O-antigen and the lipid A modification. Most of them were organized into gene clusters on chromosome 1, while the genes encoding for lipid A biosynthesis were distributed along this chromosome, except for *lptD*, which was found on chromosome 2 (Figure 6A).

An analysis of open reading frames (ORFs) in the KMM 8419 genome showed that neither *rjg* (=*tpiA*) nor either copy of *glpX* were located within the 5′- and 3′-continued boundaries of *gmhD* (=*waaD*, *rfaD*). Upstream of *gmhD* (ORF_342), there was a biosynthetic gene cluster (*wav*/*waa*) of 16 genes involved in the core LPS biosynthesis (Figure 6A). Downstream of *gmhD*, there was a 95 kb locus containing polysaccharide-related genes, which was extended to *ugd* (ORF_411, UDP-Glc 6-dehydrogenase, EC 1.1.1.22) (Figure 6B, Appendix A). Downstream of the locus, at a distance of about 900 kb, there were genes for EPS synthesis (*sypABCDGHLMNOIQR*) co-located with the genes for a lipid A modification system (*arnBCADTEF*) (Figure 6A). Based on these different transcription directions and predicted gene functions, the 68 ORFs of the 95 kb locus can be assigned to five regions (Figure 6(Ba), Appendix A). Three JUMPstart elements were found neighboring ORF_358, ORF_396 and ORF_402 (Figure 6B). The first element may regulate genes in region 1, whereas the second one regulates genes in region 4.

The genes for the nucleotide sugar biosynthesis of D-Glc and L-Rha in the CPS unit were identified in regions 4 and 5. *galF* (ORF_374) encodes a UTP-Glc-1-phosphate uridylyltransferase for UDP-Glc biosynthesis. The three genes *rmlA* (ORF_388), *rmlD* (ORF_389) and *rmlC* (ORF_392), which are responsible for the biosynthesis of dTDP-Rha, were located on the leading strand. Only *rmlB* (ORF_398) was found in region 5. RmlB catalyzes the formation of dTDP-4-keto-6-deoxy-D-Glc in the second step of L-Rha biosynthesis. Because *rmlB* is located on the lagging strand, it can affect the biosynthesis of the nucleotide-activated L-Rha. There are several GTs in the cluster that can determine the assembly of monosaccharides into the linear polysaccharide (Table 2). The GTs positioned in the same direction as other CPS-related genes belonged to the following GT families: GT2 (WbbL, RfbF, RgpF, BcbE), GT61 (DUF563), GT4 (WecA) and GT1+GT4 (RfaB).

Genes for a polysaccharide export were found in regions 1 (*kpsT*, ORF_353), 4 (*kpsD* (ORF_395), *kpsM* (ORF_394) and *kpsE* (ORF_393)), and 5 (*kpsS*, ORF_405). It is worth noting that *kpsT* was co-directed with *kpsDME*. The KpsDME proteins form a channel through outer and inner membranes, and KpsT, a peripheral inner membrane protein, hydrolyzes ATP for substrate transport [24].

Importantly, the second copies of *cysC* encoding for adenylylsulfate kinase (EC 2.7.1.25) and *cysN* and *cysD* encoding for ATP sulfurylase (EC 2.7.4.4) were found in region 1. These enzymes provide the biosynthesis of PAPS (3′-phosphoadenosine 5′-phosphosulfate), which serves as a universal sulfate donor and is transferred to various substrates by sulfotransferases. Based on eggNOG and antiSMASH annotations, three regions (2, 4 and 5) included sulfrotransferase genes (ORF_362, ORF_386 and ORF_404). The presence of ORF_386 near the rhamnose operon indicates the possibility of rhamnose sulfation.

Thus, the gene cluster identified in the *Vibrio* sp. KMM 8419 genome encodes the complete pathway for the nucleotide sugar biosynthesis, assembly, transport and sulfation of the CPS.

### 2.4. Proposed CPS Biosynthesis and Inter-Strain Sequence Analysis

Based on the above results, we suggest the following pathway for CPS assembly in the strain KMM 8419 (Figure 6C). The first step starts by WecA transferring GlcNAc to und-P. Then, WbbL adds the first L-Rha residue from dTDP-Rha to GlcNAc, and the second L-Rha residue is added by RfbF [25,26]. It is possible that RgpF, annotated as Rhamnan synthesis protein F, transfers D-Glc from UDP-Glc to the last L-Rha residue [27]. The following L-Rha residues are likely included in the growing polysaccharide already being sulfated. RfaB, annotated as GT1+GT4 with SQD2-domen, transfers the modified L-Rha to the D-Glc residue. BcbE is proposed to add the D-Glc residue to the sulfated L-Rha. Consequently, the process of adding monosaccharide units continues until the CPS reaches the required size, after which it is transported to the outer surface of the cell by the ABC transporter. Unfortunately, the exact functions of the glycosyltransferase and sulfotransferase genes are unknown and require further biochemical and genetic studies. Furthermore, in the established structure of the CPS, the L-Rha residue exhibited an O-acyl group at the C-4 position. The acyltransferase (ORF_397) responsible for transferring the acyl group was proposed to exist at the start of region 5. However, it is difficult to predict at what point the transfer of the acyl group to L-Rha occurs.

To assess the versatility of the *Vibrio* sp. KMM 8419-like CPS gene cluster, an NCBI BLAST search was conducted on GenBank (accessed on 21 November 2024 [28]) using genes for PAPS biosynthesis, GTs and STs. As a result, we failed to find publicly available genomes sharing the same or similar organizations as the CPS locus.

In addition, we performed an analysis of unpublished genomic sequences (obtained from our group) from polychaete-derived *Vibrio* spp. strains [21]. In the results, two genomes were selected for further CPS gene cluster analysis (Table 3). Given the 95–96% ANI (average nucleotide identity) and 70% dDDH (digital DNA–DNA hybridization) thresholds used to define species delineation, the strain KMM 8420, like *Vibrio* sp. KMM 8419, belongs to the same species, while the closely related strain KMM 8434 may belong to another new species (Table 3). According to the phylogenomic tree, these strains cluster together and form two distinct lineages with strong bootstrap support within the genus *Vibrio* (Appendix A).

Based on antiSMASH annotation, the CPS loci were extracted from these two genomes using *gmhD* and *rraA*/*menA* as CPS gene cluster boundaries established for KMM 8419 (Appendix A). Amazingly, the organization of the CPS gene cluster in KMM 8419 was very similar to that of KMM 8434 but was completely different from the cluster of KMM 8420, which belongs to the same species (Table 3, Appendix A). The main difference between the two CPS clusters was in region 4. Region 4 of KMM 8434 additionally contains genes encoding for nucleotide galactose and fucose biosynthesis and does not contain a sulfotransferase gene (Figure 7). This suggests a different monosaccharide composition for a non-sulfated CPS repeating unit. Surprisingly, in spite of the presence of the Rha-operon in the three genomes, the *rmlABCD* genes between KMM 8419 and KMM 8420 were significantly different (41–72% of identity). In addition, polysaccharide assembly and transport in KMM 8420 can be accomplished in a different way.

It should be noted that the structure for the O-antigen of *Vibrio* sp. KMM 8419 (as well as for those of KMM 8420 and KMM 8434) is still unknown. However, O-antigen-related genes were found downstream of *rmlB* (ORF_398) (Figure 6B, Appendix A). It is interesting that a third JUMPstart element found upstream of *wzx* (ORF_402) may regulate O-antigen biosynthesis. Based on an extensive bioinformatic analysis, the gene set includes nucleotide sugar biosynthesis genes (ORF_411, ORF_410, ORF_408); GT genes (ORF_409, ORF_407, ORF_406, ORF_403) and an O-antigen transport gene (ORF_402). Also, this operon contains *epsL* (ORF_409) encoding for an initiating GT, which may transfer GalNAc onto a lipid carrier, and *ctgA* (ORF_403), which is responsible for adding a second GalNAc residue. Furthermore, the presence of the carbohydrate sulfotransferase gene (ORF_404) indicates the potential for the sulfation of the O-antigen. The assembly of the oligosaccharide is presumably carried out in the cytoplasm, after which it is translocated to the periplasm by a flippase (Wzx, ORF_402), where it is subsequently attached to the core oligosaccharide of the LPS via O-ligase (ORF_2842). The genomic organization of the putative O-antigen gene clusters from the strains KMM 8419 and KMM 8434 was identical and completely different from that of KMM 8420 (Figure 7). Notably, the genes encoding *W*zy polymerase were absent in these genomes.

Thus, the 95 kb locus in the *Vibrio* sp. KMM 8419 genome contains genes for both CPS and O-antigen biosynthesis and encodes the complete pathways for the nucleotide sugar biosynthesis, assembly, transport and sulfation of both polysaccharides.

## 3. Discussion

Polysaccharides are promising natural polymers due to their biomedical and nutraceutical properties, mainly comprising their anticancer, antioxidant, antiviral, antibacterial, hypolipidemic, hypoglycemic and immune- and energy-regulating activities [29,30,31]. Sulfated polysaccharides are widely distributed in higher organisms. Moreover, the two most important sulfated polysaccharides, heparin and chondroitin sulfate, are found exclusively in animals. While heparin is primarily applied as an anticoagulant drug, chondroitin sulfate is widely used as an anti-inflammatory, immunomodulatory, anti-thrombotic and anticoagulant agent. Furthermore, chondroitin sulfate can be used for tissue engineering and drug delivery systems [32]. Recently, the heterologous production of sulfated chondroitin was successfully carried out in genetically engineered *Escherichia coli* [33] and *Pichia pastoris* [34]. This biosynthesis using microbial cell factories signifies significant progress in the production of other sulfated polysaccharides [35].

Sulfated polysaccharides are also found in some microorganisms, mainly marine Gram-negative bacteria. It is noteworthy that sulfated polysaccharides isolated from marine bacteria exhibit cytotoxicity and growth-inhibitory effects on human cancer cells [9,36,37,38]. Therefore, marine sulfated polysaccharides, due to their antiproliferative effects on cancer cells, are a highly promising research topic in biomedicine [39,40].

The study of genetic cluster structure becomes essential not only for obtaining insight into the fundamental basis of the diversity of polysaccharide structures but also for their biotechnological production. The genetic organization of gene clusters encoding the biosynthesis of CPS in *Vibrio* spp. has been previously described [41,42,43,44]. These studies have identified CPS gene cluster borders from *gmhD* (encoding ADP-L-glycero-D-manno-heptose-6-epimerase, EC 5.1.3.20) to *rjg* (encoding triosephosphate isomerase, EC 5.3.1.1) or *glpX* (encoding fructose-1,6-bisphosphatase, EC 3.1.3.11) in most strains of *V. cholerae* and *V. parahaemolyticus*. However, in the genomes of KMM 8419, KMM 8434 and KMM 8420, the boundaries of the CPS gene cluster were expanded to *rraA*/*menA*, probably due to the inclusion of O-antigen biosynthesis genes. A similar structural organization has previously been described for other *Vibrio* species, namely a CPS/O-antigen locus [41,44].

The CPS gene locus of the strain KMM 8419 contains genes encoding the biosynthesis of ATP-binding cassette (ABC) transporter (*kpsDME, kpsT* and *kpsS*), which is common for the synthesis of group 2 and 3 polysaccharide capsules [45]. Despite the presence of *yjbFGH* (paralog *gfcABCD*) in region 1, which can be involved in the formation of the group 4 polysaccharide capsule [46,47], all genes encoding *wzy*-dependent polysaccharide synthesis, except for *wzx* (ORF_402), were absent. Although the CPS cluster contained *kpsS*, which is responsible for the addition of the first Kdo residue to a lipid carrier, the remaining Kps-encoding genes, providing the synthesis of the group 2 polysaccharide capsule, were either absent (*kpsC*) or located far away from the cluster (*kpsF* and *kpsU*). Moreover, there was *wecA* (GT1, ORF_347) encoding for an initiating GT (EC 2.7.8.33), which can mediate the transfer of a phosphosugar from NDP-sugar to undecaprenyl phosphate (Und-P) [48]. The above suggests that the CPS of the strain KMM 8419 belongs to group 3 of the polysaccharide capsule and occurs via an undecaprenyl phosphate-dependent pathway using the ABC transporter [49,50].

A common feature of polysaccharide gene clusters is that the GC content is distinct from the rest of the chromosomal DNA, suggesting the acquisition of some regions by horizontal gene transfer (HGT). The GC content of the CPS gene cluster of KMM 8419 was 43.05 mol%, with a minimum value of 32.94 moL% (ORF_385, a predicted AB-hydrolase), while the genomic GC content was estimated to be 46 mol%. Based on the Alien Hunter results, some genes in the locus may have arisen through HGT events (Figure 6(Bb)). Notably, the closest homologues of WbbL, RfbF, RgpF and DUF563 came from the genus *Vibrio*, while the closest homologues of RfaB and BcbE came from various genera of *Proteobacteria*. In addition, the BLAST analysis showed that some genes subjected to HGT were highly similar (up to 87.12%) to those from relatives of the genera *Psychrobacter* and *Pseudoalteromonas*. It is interesting that the sulfotransferases were homologous to those from the orders *Alteromonadales* (ORF_362, ORF_386) and *Nevskiales* (ORF_404). Furthermore, the CPS gene clusters of KMM 8419 and KMM 8420 contained mobile genetic elements that might mediate HGT for environmental adaptation.

The biotechnological production of sulfated polysaccharides consists of polysaccharide synthesis, PAPS synthesis and regeneration, and the sulfation of polysaccharides by unique sulfotransferases. The CPS gene clusters KMM 8419 and KMM 8434 included genes for PAPS synthesis (*cysNDC*) and sulfotransferases. These enzymes facilitate the synthesis of the universal sulfate donor [51,52] and the transfer of the sulfated group from PAPS to the substrate. Currently, the main focus of research is on sulfotransferases from pathogenic bacteria that transfer sulfate to glycosaminoglycans. Such polysaccharides help pathogens evade the immune response because host cells have similar molecules [53]. In contrast, relatively little is known about sulfotransferases that transfer a sulfate group to the sugar residues of polysaccharides. Considering that the inclusion of a sulfate group in the polysaccharide chain changes its biological activity, polysaccharide sulfotransferases are of great interest from a biotechnological point of view [54].

## 4. Conclusions

In summary, we established the complete structure of a CPS from *Vibrio* sp. KMM 8419, isolated from a food-net mucus sample of marine polychaete *C. cautus*. The polysaccharide was characterized by chemical and spectroscopic methods. It is a novel sulfated glycan, and its disaccharide repeating unit has a unique among known bacterial polysaccharides structure (http://csdb.glycoscience.ru/bacterial (accessed on 6 November 2024) [55,56]). Whole-genome sequencing of *Vibrio* sp. KMM 8419 was carried out to deduce information about the genes involved in the biosynthesis of the sulfated CPS. It was found that the *Vibrio* sp. KMM 8419 genome contains a 95 kb polysaccharide-related locus encoding the complete pathways for the nucleotide sugar biosynthesis, assembly, transport and sulfation of both CPS and O-antigen polysaccharides. A hypothetical scheme for CPS assembly and transport was proposed. Despite the similar positioning of genes within CPS gene clusters in bacteria, the precise mechanism of CPS assembly remains unknown due to a lack of polysaccharide structure data.

Here, we report that the presence of both PAPS synthesis genes and sulfotransferase genes in the polysaccharide biosynthetic gene cluster is a hallmark of sulfated polysaccharide biosynthesis. Given the potential importance of polysaccharide sulfation for various biotechnological and biomedical applications, the genome sequencing and prediction of biosynthetic gene clusters encoding specialized enzymes (sulfotransferases and glycosyltransferases) can greatly facilitate the search for microbial producers of sulfated polysaccharides and support further studies on their biosynthesis.

## 5. Materials and Methods

### 5.1. Isolation and Purification of CPS

Strain KMM 8419 (=CB1-14) was obtained from the Collection of Marine Microorganisms (KMM) at the G.B. Elyakov Pacific Institute of Bioorganic Chemistry, Far East Branch of Russian Academy of Sciences (Vladivostok, Russia). Bacteria were isolated from a secreted mucous net of the marine polychaete *Chaetopterus cautus* (formally *Chaetopterus variopedatus*), collected at a depth of 5–8 m (salinity 33%, temperature 20 °C) at the Marine Experimental Station of the Pacific Institute of Bioorganic Chemistry, Troitza Bay, Peter the Great, the Sea of Japan, in August 2016. After primary isolation and purification, the strain was stored at −80 °C in marine broth (Difco, Sparks, MD, USA) supplemented with 20% (*v*/*v*) glycerol. The bacterium was cultivated for 48 h at ambient temperature on a medium (12 L) consisting of 5.0 g of bactopeptone, 1.0 g of yeast extract, 1.0 g of glucose, 0.2 g of K_2_HPO_4_, 0.05 g of MgSO_4_ and 750 mL of natural seawater/250 mL of distilled water at pH 7.8. Dry bacterial cells (6.9 g) were suspended in 100 mL of extraction buffer (0.22 M NaCl, 0.026 M MgCl_2_, 0.01 M KCl) and stirred continuously for 16 h at 4 °C. The cell pellet was collected by centrifugation (5000 rpm, 25 min, 4 °C), and the extraction procedure was repeated two more times (4 °C, 1 h). Supernatants were combined, dialyzed (MWCO 12,000 Da) and lyophilized to yield crude CPS (167 mg). The freeze-dried material was resuspended in 6 mL of digestion buffer (pH 7.5) containing 0.01 M TRIS and 0.01 M MgCl_2_. After the addition of 2 mg of proteinase K (Sigma, St. Louis, MO, USA), the solution was incubated for 2 h at 60 °C. After dialysis against distilled water (MWCO 12,000 Da), it was lyophilized to give rise to enzymatically treated CPS (65 mg). The freeze-dried supernatant was subjected to anion exchange chromatography on a column (10 × 1.5 cm) of Toyopearl DEAE-650M (Tosoh Bioscience, Tokyo, Japan) in a step-wise gradient of NaCl. The resulting main fraction (28.3 mg, 0.5 M NaCl) was dialyzed and freeze-dried. After that, the CPS was purified by hydrophobic interaction chromatography using a Butyl Sepharose 4FF column (10 × 1.5 cm, Sigma, St. Louis, MO, USA) with the method described in [57]. The CPS was eluted with buffer in a non-bound fraction. Finally, the CPS was desalted on the Toyopearl HW-40 column (120 × 1.5 cm, Tosoh Bioscience, Tokyo, Japan) eluted with aq 0.3% AcOH, yielding the final CPS (21 mg). Elution was monitored with a differential refractometer (Knauer, Berlin, Germany).

### 5.2. Molecular Weight Determination and Electrophoretic Analysis of the CPS

The molecular weight of the CPS was analyzed using HPLC (Agilent 1100 Series, Hamburg, Germany) equipped with successively connected columns of Shodex Asahipak GS-520 HQ and GS-620 HQ (7.5 mm × 300 mm) at 50 °C with elution by 0.15 M NaCl (0.4 mL/min). Columns were calibrated using standard dextrans of 4–400 kDa (Sigma, St. Louis, MO, USA). The electrophoresis of the CPS was performed as described in [11].

### 5.3. Compositional Analysis of CPS Samples

Monosaccharides were analyzed as acetylated methyl glycosides using appropriate authentic samples as a reference. The methanolysis of the CPS was performed as already described in [58]. The absolute configurations of sugar residues were determined through GC-MS of the acetylated (*S*)-2-octyl glycosides, as described in [59]. All the derivatives were analyzed using a Hewlett Packard 5890 chromatograph (Conquer Scientific, Poway, CA, USA) equipped with a Hewlett Packard 5973 mass spectrometer and a HP-5MS capillary column. Acetylated methyl glycosides were analyzed using the following temperature program: 150 °C for 3 min, 150 °C → 250 °C at 3 °C/min^−1^ and 250 °C for 10 min. For the acetylated (*S*)-2-octyl glycosides, the analysis was performed at 160 °C for 3 min, 160 °C → 290 °C at 3 °C/min^−1^ and 290 °C for 10 min. Fatty acid analysis was performed through GC of the methyl derivatives after methanolysis of the CPS with 2 M acetylchloride in methanol (120 °C, 4 h). Proteins were analyzed by the conventional method [60].

### 5.4. NMR Spectroscopy

^1^H and ^13^C NMR spectra of the CPS were recorded on a Bruker Avance-III (700.13 MHz for ^1^H and 176.04 MHz for ^13^C) spectrometer (Bruker, Karlsruhe, Germany) at 37 °C using acetone (δ_C_ 31.45, δ_H_ 2.225) as an internal standard. The ^1^H,^1^H-TOCSY and ^1^H,^1^H-ROESY spectra were recorded with a 180 ms duration of MLEV-17 spin-locking and a 200 ms mixing time, respectively. ^1^H,^13^C-HMBC was optimized for an 8 Hz long-range constant. The NMR study was carried out with the recommendations described in [61]. The FT-IR spectrum of the CPS was registered in a KBr pellet on a Vector 22 Fourier-transform spectrophotometer (Bruker, Karlsruhe, Germany) with a 4 cm^−1^ resolution.

### 5.5. Whole-Genome Sequencing, Assembly and Annotation

Genomic DNA was extracted from the strain KMM 8419 using the NucleoSpin Tissue kit (Macherey–Nagel, Düren, Germany), and the nanopore library was prepared using the SQK-RAD002 kit (Oxford Nanopore Technologies, Oxford, UK) and sequenced on MinION (Oxford Nanopore Technologies, Oxford, UK). Genome assembly was performed using Flye, version 2.9 [62], with the default parameters. Genome completeness and contamination were estimated by CheckM version 1.1.3 [63]. Genome annotation was carried out using the NCBI Prokaryotic Genome Annotation Pipeline (PGAP) [64], Rapid Annotation using Subsystem Technology (RAST) [65], Prokka [66], eggNOG [67], InterPro [68] and UniProt [69]. The annotation of secondary metabolite biosynthetic gene clusters was conducted using antiSMASH server version 7.0 [70]. The circular genome of the strain KMM 8419 and the CPS gene cluster structure were visualized using the Proksee platform [23]. Putative horizontal gene transfer (HGT) events were detected via the Alien Hunter tool [71] on the Proksee platform. Comparisons of the ANI and dDDH values of the strains KMM 8419, KMM 8420 and KMM 8434 with their closest neighbors were performed with the online server EzBioCloud [72] and the TYGS platform [73], respectively. A pairwise comparison between the CPS gene clusters of the strains KMM 8419, KMM 8420 and KMM 8434 was carried out using BLASTn (BLAST version 2.16.0+) run in EasyFig (version 2.2.5) [74].

## Figures and Tables

**Figure 1 ijms-25-12927-f001:**
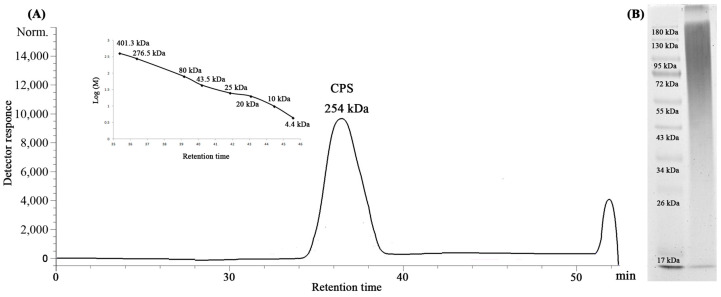
HPSEC elution profile (**A**) and toluidine blue-stained electropherogram (**B**) of the CPS from *Vibrio* sp. KMM 8419. Standard calibration curve is shown as an insertion.

**Figure 2 ijms-25-12927-f002:**
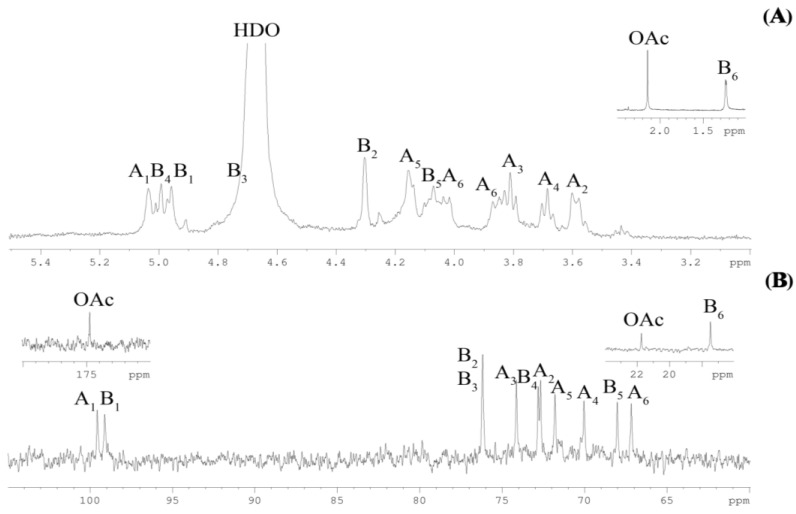
^1^H NMR spectrum (**A**) and ^13^C NMR spectrum (**B**) of the CPS from *Vibrio* sp. KMM 8419. Numerals refer to carbons and protons in sugar residues denoted by capital letters as described in Table 1.

**Figure 3 ijms-25-12927-f003:**
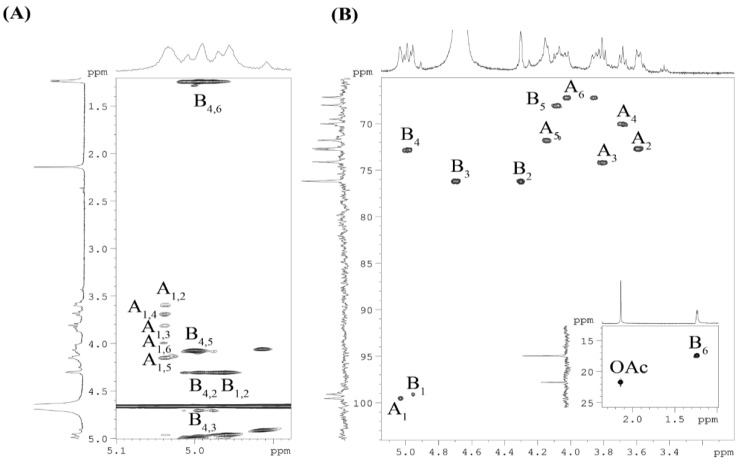
Fragments of ^1^H,^1^H TOCSY spectrum (**A**) and ^1^H, ^13^C HSQC spectrum (**B**) of the CPS from *Vibrio* sp. KMM 8419. Numerals refer to carbons and protons in sugar residues denoted by capital letters as described in Table 1.

**Figure 4 ijms-25-12927-f004:**
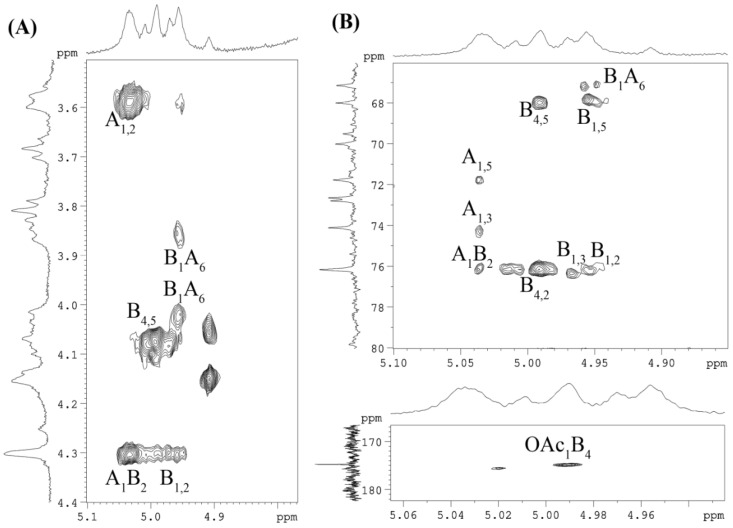
Fragments of ^1^H,^1^H ROESY spectrum (**A**) and ^1^H, ^13^C HMBC spectrum (**B**) of the CPS from *Vibrio* sp. KMM 8419. Numerals refer to carbons and protons in sugar residues denoted by capital letters as described in Table 1.

**Figure 5 ijms-25-12927-f005:**
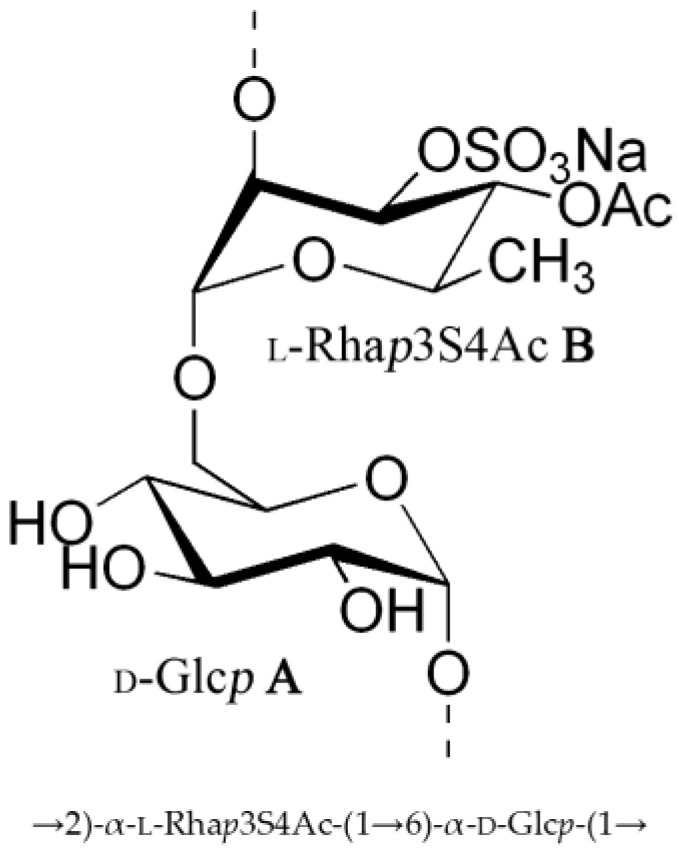
Structure of the repeating unit of the CPS from *Vibrio* sp. KMM 8419.

**Figure 6 ijms-25-12927-f006:**
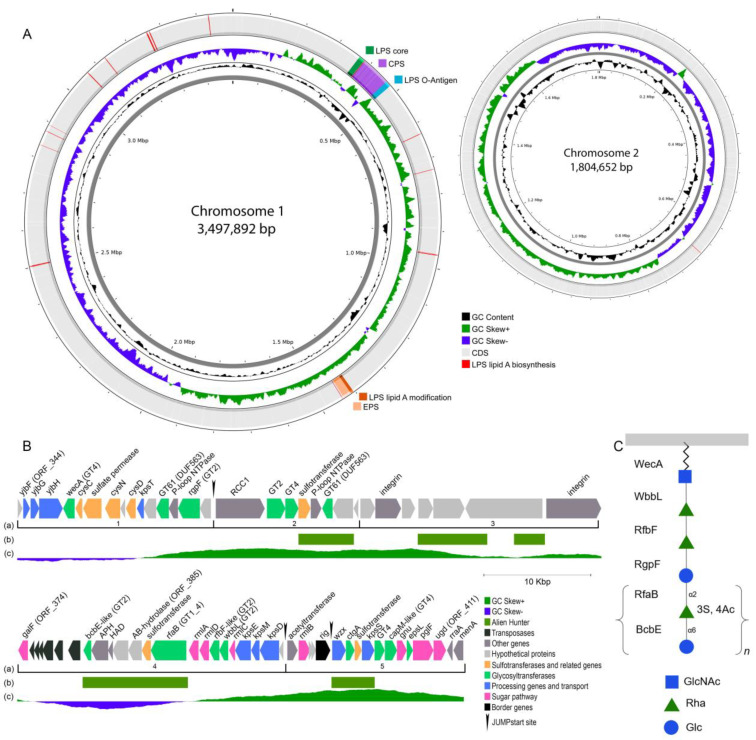
Chromosome location (**A**), gene cluster structure (**B**) and hypothetical scheme (**C**) for the CPS biosynthesis of *Vibrio* sp. KMM 8419. Chromosome and cluster visualization was performed on the Proksee server [23]. The scale is shown in megabases (Mbp) for chromosomes and in kilobases (Kbp) for gene clusters. (**a**) The numeration of regions based on the directions and functions of the ORFs is shown in Arabic numbers; (**b**) HGT regions detected by Alien Hunter tool are highlighted in green; (**c**) GC skew plot (G − C)/(G + C) is shown in violet blue and light green.

**Figure 7 ijms-25-12927-f007:**
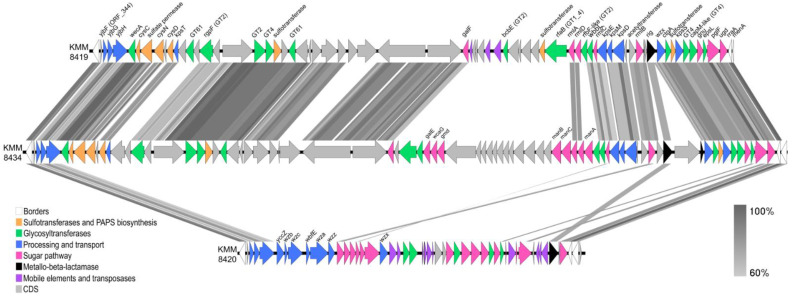
Genomic organization and genetic relatedness of CPS/O-antigen loci between *Vibrio* spp. strains. BLAST identity is shown by gray level from 100% to 60%.

**Table 1 ijms-25-12927-t001:** ^1^H and ^13^C NMR data for the CPS from *Vibrio* sp. KMM 8419; *δ*, ppm.

Sugar Residue	H-1 C-1	H-2 C-2	H-3 C-3	H-4 C-4	H-5 C-5	H-6a,b C-6
→6)-α-D-Glcp-(1→	5.04	3.59	3.80	3.68	4.15	4.02, 3.86
A	99.6	72.7	74.1	70.0	71.8	67.1
→2)-α-L-Rhap3S4Ac-(1→	4.96	4.30	4.69	4.99	4.08	1.24
B	99.1	76.2	76.2	72.8	68.0	17.5

The chemical shifts in the *O*-acetyl group are δ_H_ 2.14 and δC 21.7 (CH_3_) and 174.8 (CO).

**Table 2 ijms-25-12927-t002:** List of GT genes of *Vibrio* sp. KMM 8419 involved the CPS assembly.

ORF	Genomic Locus	Gene Name	Length bp (aa)	Product Name, EC	Family and Domain Databases
ORF_347	PG915_01855	*wecA*	1068 (355)	Undecaprenyl-phosphate alpha-N-acetylglucosaminyl 1-phosphate transferase, EC:2.7.8.33	IPR012750, ECA_WecA-rel; IPR000715, Glycosyl_transferase_4
ORF_355	PG915_01895	GT61 (DUF563)	1170 (389)	Glycosyltransferase family 61 protein	IPR049625 Glycosyltransferase 61, catalytic domain
ORF_357	PG915_01905	*rgpF*	2034 (677)	HAD-IA family hydrolase; Rhamnan synthesis protein F	IPR007739 Rhamnan synthesis F IPR036412 HAD-like superfamily
ORF_381	PG915_02025	*bcbE*-like (GT2)	726 (241)	Capsular biosynthesis protein	IPR016873 Capsular polysaccharide biosynthesis protein, BcbE, predicted IPR029044 Nucleotide-diphospho-sugar transferases
ORF_387	PG915_02055	*rfaB* (GT4)	3321 (1106)	Glycosyl transferase family 1_4 protein	IPR028098 Glycosyltransferase subfamily 4-like, N-terminal domain IPR001296 Glycosyl transferase, family 1 Sulfoquinovosyl transferase
ORF_390	PG915_02070	*rfbF* (GT2)	930 (309)	Glycosyltransferase family 2 protein	IPR029044 Nucleotide-diphospho-sugar transferases PTHR43179 RHAMNOSYLTRANSFERASE WBBL
ORF_391	PG915_02075	*wbbL* (GT2)	831 (276)	Glycosyltransferase family 2 protein	IPR029044 Nucleotide-diphospho-sugar transferases PTHR43179 RHAMNOSYLTRANSFERASE WBBL

**Table 3 ijms-25-12927-t003:** Genomic characteristics of *Vibrio* spp. strains.

Strains	Assembly Level	Genome Size (bp)	Number of Contigs	orthoANIb	dDDH	CPS/O-Antigen Length (bp)	Number of CDSs in Cluster
KMM 8419 (CB1-14)	chromosome	5,543,559	3	100	100	97,233	80
KMM 8434 (CB2-8)	contigs	5,539,082	8	95.91	67.9	114,585	72
KMM 8420 (CB2-10)	chromosome	5,536,255	3	99.64	94.4	51,079	50

## Data Availability

The GenBank accession numbers for chromosomes 1 and 2 of strain CB1-14 (=KMM 8419) are CP115920.1 and CP115921.1, respectively. The GenBank/RefSeq assembly accessions are GCA_040412085.2 and GCF_040412085.2, respectively.

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
