# Peer review of "Structure and Biosynthetic Gene Cluster of Sulfated Capsular Polysaccharide from the Marine Bacterium Vibrio sp. KMM 8419"

_ijms, 2024, doi:10.3390/ijms252312927_

Round 1
Reviewer 1 Report
Comments and Suggestions for Authors
The authors have worked and proposed a molecular approach for modelling genes of sulfated capsular polysaccharide for Vibrio bacteria of marine origin.
This work is interesting and can advance to improvements for subsequent final evaluation.
In the Introduction, the final paragraph must be restructured. First, the objectives must be defined clearly and concisely. Second, the advantages of this approach over previous relevant works should be described. That way, readers can have a clear overview of the study.
Procedures
The authors have used only one strain for their studies, and this is a significant disadvantage. The authors must repeat the work using a second organism, in order to compare the results. Will the model yield similar results or else this would provide differences? If results are similar, the model is fully validated. In case of differences, these differences must be explained.
Also, please provide a new sub-section where you would describe clearly all the control consumables and procedures that you used and took during the study.
Results
Personally, I disagree with the approach to join results and discussion in a common section. In this respect, I urge the authors to separate the two sections.
Moreover, greater use of tables should be performed to show the results, at the same time reducing the amount of text in the manuscript.
The figures in the manuscript are very good, well-done.
References. For such a complex study that develops over several scientific fields, I believe that the number of references is small. I shall expect 70 to 75 references in the revised version, which also indicates that further ideas must be developed in the revised manuscript.
This is a problem that arises from merging Results and Discussion and limits the ideas that can be developed. Separation of the two sections will also enhance the content of the manuscript.
Conclusions.
This section is very long. Some of the passages in there can be transferred to the new Discussion.
The section must be brief and must also convey a strong ‘take-home’ message for future readers.
Overall.
Revision and re-evaluation.
The issue of using a second strain in the study is paramount for acceptance.
Author Response
Reviewer 1. Comments and Suggestions for Authors
The authors have worked and proposed a molecular approach for modelling genes of sulfated capsular polysaccharide for Vibrio bacteria of marine origin. This work is interesting and can advance to improvements for subsequent final evaluation.
Response: Thank you very much for taking the time to review our manuscript. We appreciate your valuable comments and suggestions. We have carefully revised our manuscript in accordance with your comments.
Comment 1: In the Introduction, the final paragraph must be restructured. First, the objectives must be defined clearly and concisely. Second, the advantages of this approach over previous relevant works should be described. That way, readers can have a clear overview of the study.
Response: We have taken into account your proposal to reconstruct the last paragraph of the introduction. We have specified the goal and approaches to achieve it (Lines 75-77).
Comment 2: Procedures. The authors have used only one strain for their studies, and this is a significant disadvantage. The authors must repeat the work using a second organism, in order to compare the results. Will the model yield similar results or else this would provide differences? If results are similar, the model is fully validated. In case of differences, these differences must be explained. Also, please provide a new sub-section where you would describe clearly all the control consumables and procedures that you used and took during the study.
Response: We hope that we have understood your question correctly. In this work we used well-established bioinformatic tools such as antiSMASH, Blast, InterPro, EggNOG and so on (Look at, please, Materials and Methods, Lines 440-451). The tools and databases used by the servers are specified in the corresponding articles cited. No additional tool was used to search for sulfotransferases. Our approach to searching for a cluster of genes for the biosynthesis of sulfated polysaccharide was the presence of genes responsible for the biosynthesis of PAPS and genes encoding sulfotransferases in the cluster. We also added the results of the analysis of two other CPS/ O-antigen clusters (2.4. Proposed CPS biosynthesis and inter-strain sequence analysis, Lines 238-265). This approach allowed us to describe the new gene cluster for another sulfated CPS polysaccharide from Cobetia marina (unpublished data).
Comment 3: Results. Personally, I disagree with the approach to join results and discussion in a common section. In this respect, I urge the authors to separate the two sections. Moreover, greater use of tables should be performed to show the results, at the same time reducing the amount of text in the manuscript.
Response: The results and discussion are divided into two separate sections. We hope that the presented material will be more understandable to future readers.
Comment 4: The figures in the manuscript are very good, well-done.
Response: Thank you very much.
Comment 5: References. For such a complex study that develops over several scientific fields, I believe that the number of references is small. I shall expect 70 to 75 references in the revised version, which also indicates that further ideas must be developed in the revised manuscript. This is a problem that arises from merging Results and Discussion and limits the ideas that can be developed. Separation of the two sections will also enhance the content of the manuscript.
Response: We created a separate discussion section and strengthened the discussion of the results obtained by highlighting several areas of polysaccharide research (Lines 286-354).
Comment 5: Conclusions. This section is very long. Some of the passages in there can be transferred to the new Discussion. The section must be brief and must also convey a strong ‘take-home’ message for future readers.
Response: We have shortened the text of the Conclusion, leaving for us the most important results, problems and solutions.

Reviewer 2 Report
Comments and Suggestions for Authors
The manuscript titled “Structure and biosynthetic gene cluster of sulfated capsular polysaccharide from the marine bacterium Vibrio sp. KMM 8419” by Kokoulin, M.S.; et al. is a scientifi work where the authors characterize the gene cluster of capsular polysaccharide of Vibrio sp. KMM 8419 and its structure. For it, many complementary techniques were devoted in this research. The manuscript is generally well-written and this is a topic that could be interesting for a certain specialized audience in this field.
However, it exists some points that need to be addressed (please, see them below detailed point-by-point) to improve the scientific quality of the submitted manuscript paper before this article will be consider for its publication in the International Journal of Molecular Sciences.
1) The authors should consider to modify the term “Vibrio” by “Vibrio sp. KMM 8419” in the keyword list
2) “The main interest in the study of polysaccharides (…) various industrial sectors, including pharmaceuticals” (lines 38-40). Could the authors provide quantitative data insights according to the worldwide economic impact in the pharmaceutical sector? This will significantly aid the potential readers to better understand the significance of the devoted research in this work.
3) “The structural versatility of bacterial polysaccharides is associated with a broad spectrum of biological functions (…) and others” (lines 46-49). Here, even if I agree with the information provided by the authors it should not be neglected how the bacterial cell membrane composition [1] can tailor their nanomechanical properties [2] which is relevant to design future trends in nanobiomedicine.
[1] Auer, G.K.; et al. Bacterial Cell Mechanics. Biochemistry 2017, 56, 3710-3724. https://doi.org/10.1021/acs.biochem.7b00346
[2] Magazzù, A.; et al. Investigation of Soft Matter Nanomechanics by Atomic Force Microscopy and Optical Tweezers: A Comprehensive Review. Nanomaterials 2023, 13, 963. https://doi.org/10.3390/nano13060963
4) “2.1. CPS extraction, purification, and general characterization” (lines 73-92). What was the extraction yield? Some information should be furnished in this regard. Then, did the authors observe any protein aggregation effect during the purification process?
5) Then, did the authors carry out some “in vivo” experiments?
6) “3. Conclusion” (lines 247-278). This section perfectly remarks the most relevant outcomes found by the authors in this work and also the future perspectives. The authors should furnish a brief statement to discuss about the future action lines to pursue the topic covered in this research.
Author Response
Reviewer 2. Comments and Suggestions for Authors
The manuscript titled “Structure and biosynthetic gene cluster of sulfated capsular polysaccharide from the marine bacterium Vibrio sp. KMM 8419” by Kokoulin, M.S.; et al. is a scientifi work where the authors characterize the gene cluster of capsular polysaccharide of Vibrio sp. KMM 8419 and its structure. For it, many complementary techniques were devoted in this research. The manuscript is generally well-written and this is a topic that could be interesting for a certain specialized audience in this field. However, it exists some points that need to be addressed (please, see them below detailed point-by-point) to improve the scientific quality of the submitted manuscript paper before this article will be consider for its publication in the International Journal of Molecular Sciences.
Response: Thank you very much for taking the time to review our manuscript. We appreciate your valuable comments and suggestions. We have carefully revised our manuscript in accordance with your comments.
Comment 1: The authors should consider to modify the term “Vibrio” by “Vibrio sp. KMM 8419” in the keyword list
Response: Thank you, it was corrected.
Comment 2: “The main interest in the study of polysaccharides (…) various industrial sectors, including pharmaceuticals” (lines 38-40). Could the authors provide quantitative data insights according to the worldwide economic impact in the pharmaceutical sector? This will significantly aid the potential readers to better understand the significance of the devoted research in this work.
Response: Thank you for your suggestions to emphasize economic impact of such kind of research in the pharmaceutical and other sectors. Based on your recommendation, we have added the most important literature data to the Introduction (Lines 40-41) and Discussion (Lines 287-303).
Comment 3: “The structural versatility of bacterial polysaccharides is associated with a broad spectrum of biological functions (…) and others” (lines 46-49). Here, even if I agree with the information provided by the authors it should not be neglected how the bacterial cell membrane composition [1] can tailor their nanomechanical properties [2] which is relevant to design future trends in nanobiomedicine.
[1] Auer, G.K.; et al. Bacterial Cell Mechanics. Biochemistry 2017, 56, 3710-3724. https://doi.org/10.1021/acs.biochem.7b00346
[2] Magazzù, A.; et al. Investigation of Soft Matter Nanomechanics by Atomic Force Microscopy and Optical Tweezers: A Comprehensive Review. Nanomaterials 2023, 13, 963. https://doi.org/10.3390/nano13060963
Response: Thank you for your recommendation. We have expanded the introduction to include information on adaptive modifications bacterial cell wall by alterations in the cell mechanics (Lines 55-61). Unfortunately, the proposed review is too specialized for us. Thanks for the advice, we will try to study this area of ​​research more thoroughly.
Comment 4: “2.1. CPS extraction, purification, and general characterization” (lines 73-92). What was the extraction yield? Some information should be furnished in this regard. Then, did the authors observe any protein aggregation effect during the purification process?
Response: From 12 L of medium, the yield of CPS was 21 mg (Line 405). In our case, no effect of protein aggregation was observed.
Comment 5: Then, did the authors carry out some “in vivo” experiments?
Response: In the future, we plan to conduct in vivo experiments to test the cytotoxic effects, as well as gene knockout experiments and heterologous expression of CPS biosynthetic gene clusters.
Comment 6: “3. Conclusion” (lines 247-278). This section perfectly remarks the most relevant outcomes found by the authors in this work and also the future perspectives. The authors should furnish a brief statement to discuss about the future action lines to pursue the topic covered in this research.
Response: Thank you for your advice. “A brief statement to discuss about the future action” will be “genome sequencing and prediction of biosynthetic gene clusters encoding specialized enzymes (sulfotransferases and glycosyltransferases) can greatly facilitate the search for microbial producers of sulfated polysaccharides and support further studies of their biosynthesis” (Lines 369-375).

Round 2
Reviewer 1 Report
Comments and Suggestions for Authors
The authors addressed also the issues raised previously.